# Chronic Kidney Disease Progression—A Challenge

**DOI:** 10.3390/biomedicines12102203

**Published:** 2024-09-27

**Authors:** Silvio Maringhini, Carmine Zoccali

**Affiliations:** 1Istituto Mediterraneo per i Trapianti e Terapie ad Alta Specializzazione (ISMETT), 90127 Palermo, Italy; 2Renal Research Institute, New York, NY 10065, USA; carmine.zoccali@icloud.com; 3Institute of Molecular Biology and Genetics (Biogem), 83031 Ariano Irpino, Italy; 4Associazione Ipertensione Nefrologia Trapianto Renale (IPNET), c/o Nefrologia, Grande Ospedale Metropolitano, 89124 Reggio Calabria, Italy

**Keywords:** chronic kidney disease (CKD), end-stage kidney disease (ESKD), cardiovascular complications, CKD progression

## Abstract

Chronic kidney disease (CKD) is a progressive condition characterized by a continuous decline in renal function, independent of the initial cause of damage or external factors such as infection, inflammation, or toxins. The accurate measurement of renal function, typically assessed using the glomerular filtration rate (GFR), is crucial for managing CKD. The most accepted hypothesis for CKD progression is glomerular damage caused by hyperfiltration. Various factors can accelerate CKD progression, and several biomarkers have been identified to monitor this progression. Numerous studies have explored the risk factors associated with CKD progression, and some of these factors can be modified. Additionally, several drugs are now available that can reduce CKD progression. This review summarizes recent publications and highlights potential future research directions in CKD progression. It discusses the evolution of GFR measurement methods, the mechanisms driving CKD progression, and the latest findings on biomarkers and risk factors. Furthermore, it explores therapeutic strategies, including dietary modifications and pharmacological interventions, to slow CKD progression. Understanding these mechanisms and interventions is crucial for developing effective therapeutic strategies to prevent or slow CKD progression.

## 1. Introduction

Chronic kidney disease (CKD) is a progressive condition where the reduction in renal function continues independently of the initial cause of damage or external factors such as infection, inflammation, or toxins. CKD is a significant public health issue, affecting millions of people worldwide and leading to increased morbidity and mortality. The disease is characterized by a gradual loss of kidney function over time, which can eventually lead to end-stage renal disease (ESRD) requiring dialysis or kidney transplantation. According to the Global Burden of Disease Study, CKD was the 18th leading cause of death globally in 1990 and was raised to the 9th in 2019 [1,2]. The prevalence of CKD is increasing, with an estimated 9.1% of the global population affected, translating to approximately 700 million people [3]. The economic burden of chronic kidney disease (CKD) is significant on a global scale. An analysis across 31 countries/regions projected that, between 2022 and 2027, the annual direct healthcare costs of CKD (pre-kidney replacement therapy) will increase by 8.7%, from approximately USD 202 billion to USD 220 billion [4]. The increasing prevalence of CKD is driven by the rising incidence of diabetes and hypertension, which are the leading causes of CKD. Additionally, the aging population contributes to the growing burden of CKD, as the prevalence of the disease increases with age [5]. The most accepted hypothesis for CKD progression is glomerular damage caused by hyperfiltration. Various factors can accelerate CKD progression, and several biomarkers have been identified to monitor this progression. Numerous studies have explored the risk factors associated with CKD progression, and some of these factors can be modified. Additionally, several drugs are now available that can reduce CKD progression. This paper is a narrative review of the progression of CKD. We started with a PubMed search of original papers and reviews dealing with the various aspects of CKD covered in our paper and extracted from selected papers additional references that were not retrieved in the original search. Even though this was not a systematic review, ours was an orderly, well-organized approach.

## 2. How to Measure Renal Function

The quest to accurately gauge kidney function has been a persistent undertaking in the medical community, with the glomerular filtration rate (GFR) standing as the definitive indicator of renal health. The historical trajectory of estimating GFR is marked by continuous refinement of formulas, each seeking to balance precision and accuracy with practicality in clinical settings.

### 2.1. Gold-Standard Methods for Measuring GFR

In the early days, the direct measurement of GFR involved cumbersome and intricate methods, such as inulin clearance, which, while accurate, were far too complex for routine clinical use. Subsequent gold-standard methods include contrast agents like iothalamate and iohexol, which are measured using highly precise methods like high-performance liquid chromatography (HPLC), tandem mass spectrometry (MS/MS), or enzyme-linked immunosorbent assay (ELISA). These clearance methods involve administering the contrast agent intravenously and measuring its plasma concentration over time. Radioisotope methods include I-125 iothalamate and 51Cr-EDTA, which allow simultaneous renal imaging and GFR measurement [6,7].

### 2.2. Estimates of GFR

Until the late 1970s, creatinine clearance, requiring 24 h urine collections, was the sole practical method for estimating GFR in clinical practice. The introduction of the Cockcroft–Gault (CG) formula in the 1970s simplified the creatinine clearance estimate by using serum creatinine levels along with patient age, weight, and sex [8]. The Modification of Diet in Renal Disease (MDRD) Study equation, developed in the late 1980s, performed better than measured creatinine clearance but was primarily validated in patients with CKD [9]. The Chronic Kidney Disease Epidemiology Collaboration (CKD-EPI) introduced its formula in 2009, improving GFR estimation across a broader spectrum of GFR levels and in a more diverse patient population [10]. Cystatin C, another endogenous filtration marker, is less affected by muscle mass or diet, making it a valuable alternative to creatinine-based estimates [11]. The combined use of creatinine and cystatin C in the CKD-EPI formula provides the most reliable GFR estimates for clinical practice [12]. However, the inclusion of race in these formulas has sparked debate about health equity [13], leading to the development of race-free formulas. The European Kidney Function Consortium (EKFC) developed the Full Age Spectrum (FAS) Creatinine Equation in 2021, which estimates GFR across the full age spectrum without using race as a variable [14]. The EKFC’s formula, which rescales serum cystatin C and creatinine, surpasses the corresponding CKD-EPI equations in accuracy and bias [15]. Magnetic resonance imaging (MRI) offers a promising approach for measuring glomerular filtration rate (GFR), particularly in patients with impaired renal function. MRI-based GFR measurements are considered worthwhile because they can be accurate, reproducible, and relatively easy to obtain. Studies have shown that MRI-based GFR estimates are significantly more accurate than those based on creatinine levels, with 95% of MR-based GFRs falling within 30% of reference values. Even though costly, from a purely technical standpoint, MRI is a potential tool for the follow-up and management of patients with renal impairment [16,17]. Cationized ferritin-enhanced MRI (CFE-MRI) is used to assess kidney pathology with high precision [18]. Cationic ferritin (CF) acts as a superparamagnetic contrast agent that binds to the glomerular basement membrane after intravenous injection. This allows for direct, whole-kidney measurements of glomerular number, volume, and volume distribution [19]. CFE-MRI has been applied to study the transition from acute kidney injury (AKI) to chronic kidney disease (CKD), providing insights into the heterogeneity of kidney pathology during this process. Additionally, it has been used to investigate the relationship between glomerular number and volume in different mouse models, offering valuable data for understanding congenital nephron reduction. Overall, CFE-MRI is a powerful tool for non-destructively phenotyping kidney structures and tracking pathological changes over time. Inulin clearance is a time-honored method which has virtually been abandoned for its logistic complexity and limited repeatability. This method is known for its analytical imprecision. It has a coefficient of variation (CV) for repeated inulin concentration measurements of approximately 7%. This indicates that, while it is a reliable method for measuring GFR, there is some variability in the results due to the inherent imprecision in the measurement process [20].

### 2.3. Proteinuria in CKD: The Risk for CKD Progression and Cardiovascular Events

Proteinuria, the presence of excess proteins in the urine, is a significant marker and risk factor for both the progression of chronic kidney disease (CKD) and the development of cardiovascular disease (CVD). Numerous studies have established a strong association between proteinuria and adverse renal and cardiovascular outcomes. Proteinuria is a critical predictor of CKD progression. It has been shown that higher levels of proteinuria correlate with faster rates of kidney function decline. In a mass screening study involving over 107,000 participants in Okinawa, Japan, proteinuria was identified as the most powerful predictor of end-stage renal disease (ESRD) risk over a 10-year period in the general population [21]. This finding underscores the importance of proteinuria as a prognostic marker in CKD management. The pathophysiological mechanisms linking proteinuria to CKD progression involve several factors. Proteinuria can cause direct damage to the renal tubules, leading to inflammation and fibrosis. Additionally, the presence of proteins in the urine can trigger a cascade of events that exacerbate kidney damage, including the activation of the renin–angiotensin–aldosterone system (RAAS) and increased oxidative stress [22]. Beyond its impact on kidney function, proteinuria is also a significant risk factor for cardiovascular events. Patients with CKD and proteinuria are at a higher risk of developing cardiovascular complications, such as heart failure, myocardial infarction, and stroke. The relationship between proteinuria and CVD is partly due to shared risk factors, such as hypertension and diabetes, but proteinuria itself may also contribute to cardiovascular risk through mechanisms like endothelial dysfunction and increased arterial stiffness [23].

### 2.4. Current Classification of CKD

The Kidney Disease: Improving Global Outcomes (KDIGO) 2024 Clinical Practice Guideline [24] provides a comprehensive framework for the management of CKD, including the prognostic classification of CKD based on GFR and albuminuria categories. This classification system helps clinicians assess the risk of CKD progression and guide treatment decisions. The KDIGO 2024 guideline classifies CKD into five stages based on GFR, ranging from G1 (normal or high GFR) to G5 (kidney failure). Additionally, it incorporates three categories of albuminuria (A1, A2, and A3) to further stratify risk (permission to reproduce not granted by the International Society of Nephrology). For instance, a patient with a GFR of 45–59 mL/min/1.73 m^2^ (stage G3a) and an albumin-to-creatinine ratio (ACR) of >300 mg/g (category A3) would be considered at high risk for CKD progression and cardiovascular events. In conclusion, proteinuria is a potent predictor of both CKD progression and cardiovascular events. The effective management of proteinuria, as outlined in the KDIGO 2021 guidelines, is crucial for improving outcomes in patients with CKD.

### 2.5. The Concept of Renal Reserve

Renal reserve, also known as renal functional reserve (RFR), is a physiological concept that describes the kidney’s ability to increase its glomerular filtration rate (GFR) in response to certain stimuli, such as a protein load or amino acid infusion. This concept was first introduced by Bosch and colleagues in 1983, who demonstrated that normal kidneys could significantly augment their filtration capacity when challenged with a protein load [25].

#### 2.5.1. Underlying Physiology

The baseline GFR, which is relatively stable over long periods, represents the kidney’s filtration capacity under normal, unstressed conditions. However, the kidneys possess an intrinsic ability to enhance their filtration rate when needed, a phenomenon termed renal reserve. This increase in GFR is primarily mediated by hemodynamic changes, including afferent arteriolar dilation and efferent arteriolar constriction, which increase glomerular capillary pressure and, consequently, the filtration rate [26,27]. The physiological basis for renal reserve lies in the kidney’s ability to recruit additional nephrons or enhance the function of existing nephrons. Under normal conditions, not all nephrons are operating at their maximum capacity. When a demand for increased filtration arises, such as during a high-protein meal, the kidneys can mobilize these “reserve” nephrons to increase overall GFR. This adaptive mechanism ensures that the body can handle transient increases in metabolic waste products without compromising homeostasis [27].

#### 2.5.2. Pathophysiology

The concept of renal reserve is conceptually relevant in the context of renal disease. In the early stages of chronic kidney disease (CKD), the baseline GFR may remain within normal limits, but the renal reserve is often diminished. This reduction in renal reserve can be an early indicator of subclinical kidney damage before any noticeable decline in baseline GFR occurs. As kidney disease progresses, the ability to recruit additional nephrons diminishes, leading to a reduced or absent renal reserve [28]. Acute kidney injury (AKI) can also impact renal reserve. Patients with a reduced renal reserve are more susceptible to AKI because their kidneys lack the capacity to respond to additional stressors. This means that renal reserve is a potentially valuable tool for predicting the risk of AKI in vulnerable populations [25].

#### 2.5.3. Clinical Application

Despite its physiological and pathophysiological significance, the concept of renal reserve is rarely applied in clinical practice. Several factors contribute to this. First, the assessment of renal reserve typically requires specific stimuli, such as amino acid infusion or protein loading, followed by the precise measurement of GFR. These procedures are time-consuming, costly, and not routinely available in most clinical settings [25,27]. Second, there is a lack of standardized protocols for measuring renal reserve, complicating its integration into routine clinical practice. Variability in testing methods and in the interpretation of results further limits its utility. Additionally, the clinical relevance of renal reserve in guiding treatment decisions remains uncertain, as there is limited evidence linking renal reserve measurements to improved patient outcomes [27]. Finally, nephrology has traditionally focused on baseline GFR and proteinuria as primary indicators of kidney function and disease progression. These parameters are easier to measure and have well-established prognostic value, making them more practical for routine use. In conclusion, while the concept of renal reserve provides valuable insights into kidney physiology and early disease detection, its application in clinical practice is limited by practical and methodological challenges. Further research is needed to standardize testing protocols and establish the clinical utility of renal reserve measurements in improving patient outcomes [27,29].

## 3. Prospective Studies

The Chronic Renal Insufficiency Cohort (CRIC) Study and the Chronic Kidney Disease in Children (CKiD) studies are notable examples of studies that have provided valuable insights into CKD progression and related risk factors. The CRIC Study, a multicenter, prospective cohort study in the United States, has been ongoing for over twenty years and has produced a long list of publications on CKD risk factors [30], including hypertension, diabetes, cardiovascular disease, and non-traditional risk factors. It has also highlighted the importance of early detection and management of CKD to slow its progression [31].

The CKiD study focuses on pediatric CKD and has highlighted the impact of factors such as hypertension, malnutrition, and puberty on CKD progression in children. The CKiD study has shown that children with CKD secondary to glomerular disease have a higher risk of CKD progression, compared to those with congenital abnormalities of the kidney and urinary tract [32].

A global extended database combining cohort studies in CKD, the iNET-CKD (International Network of CKD cohort studies), has been established to promote collaborative research, foster the exchange of expertise, and create opportunities for research training. Genetic, behavioral, and health services factors associated with the course of CKD can be investigated in this extensive database, creating a tremendous opportunity for international collaboration [33].

## 4. Multifactorial Origins of CKD

The etiology of CKD is inherently multifactorial, involving a complex interplay of genetic, environmental, and lifestyle factors. Genetic predispositions, such as mutations in the *APOL1* gene, can increase susceptibility to CKD, particularly in certain ethnic groups [34]. Environmental factors, including exposure to nephrotoxic agents and infections, can initiate or exacerbate kidney damage [28,35]. Lifestyle factors, such as a poor diet, a lack of exercise, and smoking, contribute to the development and progression of CKD by promoting conditions like hypertension and diabetes, which are major risk factors for CKD [36]. Additionally, socioeconomic status and access to healthcare play significant roles in the management and outcomes of CKD. Understanding the multifactorial nature of CKD is essential for developing comprehensive prevention and treatment strategies that address the diverse factors contributing to the disease [31].

## 5. Progression of Kidney Damage

Chronic kidney disease (CKD) is a progressive condition characterized by a gradual loss of kidney function over time. The progression of kidney damage is influenced by a multitude of risk factors, which can be broadly categorized into non-modifiable and modifiable factors. Different etiologies of CKD produce different rates of progression of kidney damage; congenital anomalies of the kidneys and urinary tract (CAKUT) represent up to 40–50% of childhood CKD and it has been shown that the progression of renal damage is lower here than in glomerular diseases [37]. Furthermore, in infants with kidney damage, the GFR usually increases during early childhood and then has a period of stability, before subsequently progressing to CKD in adolescence or adulthood [24]. We will examine and provide a comprehensive overview of these risk factors, biomarkers, prognostic models, and notable prospective studies, with a focus on the findings from various key studies.

### 5.1. Risk Factors for CKD Progression

Risk factors for CKD progression are diverse and can be grouped into several categories, including demographic factors, genetic factors, cardiovascular factors, metabolic factors, lifestyle factors, and others (Table 1) [31,32].

#### 5.1.1. Demographic and Genetic Factors

Non-modifiable risk factors are inherent to the individual and cannot be altered, but they play a significant role in the susceptibility and progression of CKD. Geographical location and socioeconomic status are critical determinants of health outcomes, including CKD progression. Studies have shown that individuals living in low-income areas or regions with limited access to healthcare are at a higher risk of CKD progression [35]. Additionally, race and ethnicity play a significant role, with African Americans and Hispanics being more susceptible to CKD progression than Caucasians [38,39]. This disparity is partly due to genetic factors such as the *APOL1* gene, which is more prevalent in African American populations and is associated with a higher risk of kidney disease [34,40]. Uromodulin gene variants have emerged as a possible cause of CKD progression [41]. Age and sex are also important non-modifiable risk factors. The prevalence of CKD increases with age, and men are generally at a higher risk of CKD progression compared to women [42]. Birth weight is another non-modifiable factor, with low birth weight being associated with an increased risk of CKD later in life [43].

#### 5.1.2. Cardiovascular and Metabolic Factors

Cardiovascular conditions such as hypertension, heart failure, and atrial fibrillation are critical modifiable risk factors. Hypertension, in particular, has been confirmed as a significant risk factor in several studies [44,45,46]. Hypertension contributes to CKD progression by causing damage to the blood vessels in the kidneys, leading to reduced kidney function over time. The effective management of blood pressure (BP) is crucial in slowing the progression of CKD. The lack of dipping of blood pressure at night may be a risk factor independent from high blood pressure values [47]. It is questionable whether recommended BP values should be applied to CKD patients [48]. Heart failure and atrial fibrillation are also associated with CKD progression. These conditions can lead to reduced blood flow to the kidneys, exacerbating kidney damage. Managing these cardiovascular conditions through medication and lifestyle changes can help mitigate their impact on CKD progression. Metabolic factors including fibroblast growth factor 23 (FGF23), serum bicarbonate, urinary oxalate, serum uric acid, and parathyroid hormone levels are also associated with CKD progression. Elevated levels of FGF23, for example, have been linked to increased mortality and faster progression of CKD [49]. Similarly, anemia, abnormalities in serum bicarbonate, urinary oxalate, and serum uric acid levels can contribute to kidney damage and CKD progression [32].

#### 5.1.3. Lifestyle and Behavioral Factors

Behavioral factors such as smoking, diet, physical activity, and drug use are modifiable and have a substantial impact on CKD progression. Smoking is a well-known risk factor for many chronic diseases, including CKD. It contributes to kidney damage by increasing blood pressure and reducing blood flow to the kidneys [50]. Diet and physical activity are also important modifiable risk factors. A diet high in sodium, processed foods, and unhealthy fats can contribute to hypertension and other metabolic disorders, increasing the risk of CKD progression. Conversely, a diet rich in fruits, vegetables, and whole grains can help protect kidney function [51]. Regular physical activity can also help manage blood pressure and improve overall health, reducing the risk of CKD progression [52,53]. Socioeconomic status, although often considered non-modifiable, can be influenced through policy and social interventions. Improving access to healthcare, education, and healthy food options can help mitigate the impact of socioeconomic status on CKD progression.

#### 5.1.4. Other Factors

Other significant factors include acute kidney injury (AKI), renal fibrosis, gut–renal axis interactions, inflammatory states, and pregnancy. These factors can either exacerbate existing kidney damage or contribute to the onset of CKD. Acute kidney injury (AKI) is a sudden episode of kidney failure or damage that occurs within a few hours or days. 

AKI can lead to a rapid decline in kidney function and increase the risk of CKD progression. Preventing and managing AKI through early detection and appropriate treatment is crucial in reducing its impact on CKD progression [54,55]. Renal fibrosis, the formation of scar tissue in the kidneys, is a common feature of CKD. It results from chronic inflammation and damage to the kidney tissues. Renal fibrosis can lead to a decline in kidney function and increase the risk of CKD progression. Targeting the underlying causes of renal fibrosis, such as inflammation and oxidative stress, can help slow the progression of CKD [56]. The gut–renal axis refers to the bidirectional relationship between the gut microbiota and kidney function. Dysbiosis, an imbalance in the gut microbiota, has been linked to CKD progression. Modulating the gut microbiota through diet, probiotics, and other interventions may help improve kidney function and slow CKD progression. Inflammatory states, such as chronic inflammation and autoimmune diseases, can contribute to CKD progression. Managing these conditions through medication and lifestyle changes can help reduce inflammation and protect kidney function [57]. Pregnancy is another factor that can impact CKD progression. Pregnant women with CKD are at a higher risk of complications, including preeclampsia and preterm birth. Managing CKD during pregnancy through close monitoring and appropriate treatment is crucial in reducing the risk of CKD progression and improving maternal and fetal outcomes [58]. Puberty accelerates the decline of GFR [59,60].

### 5.2. Biomarkers for CKD Progression

Several biomarkers have been identified for predicting CKD progression (Table 2). Traditional markers such as serum creatinine and albuminuria are widely accepted and used in the KDIGO classification of CKD. Serum creatinine is a waste product produced by muscle metabolism and is filtered by the kidneys. Elevated levels of serum creatinine indicate reduced kidney function. Albuminuria, the presence of albumin in the urine, is a marker of kidney damage and is associated with an increased risk of CKD progression. 

TNFR1 (tumor necrosis factor receptor 1) and TNFR2 (tumor necrosis factor receptor 2) are receptors for tumor necrosis factor-alpha (TNF-α), a pro-inflammatory cytokine. Elevated levels of TNFR1 and TNFR2 are associated with increased inflammation and kidney damage. These biomarkers have been studied in several cohorts and have shown a consistent association with CKD progression. However, their independent predictive value has not been confirmed in multiple cohort studies or randomized clinical trials; thus, they are not yet considered validated [61].

suPAR (soluble urokinase-type plasminogen activator receptor) is another biomarker of inflammation and is associated with CKD progression. Elevated levels of suPAR have been linked to an increased risk of CKD progression in several studies. However, similarly to TNFR1 and TNFR2, suPAR has not been validated in multiple cohort studies or randomized clinical trials [62].

KIM-1 (kidney injury molecule-1) is a biomarker of tubular injury and is elevated in patients with CKD. KIM-1 has been studied in various cohorts and has shown an association with CKD progression. However, its independent predictive value has not been confirmed in multiple cohort studies or randomized clinical trials, and, thus, it is not considered validated [63].

NGAL (neutrophil gelatinase-associated lipocalin) is a biomarker of acute kidney injury and is also elevated in CKD. NGAL has been studied in several cohorts and has shown an association with CKD progression. However, its independent predictive value has not been confirmed in multiple cohort studies or randomized clinical trials, and, thus, it is not considered validated [64].

L-FABP (liver-type fatty acid-binding protein) is a biomarker of tubular injury and is associated with CKD progression. L-FABP has been studied in various cohorts and has shown an association with CKD progression. However, its independent predictive value has not been confirmed in multiple cohort studies or randomized clinical trials, and, thus, it is not considered validated [65].

UMOD (uromodulin) is a protein produced by the kidneys and is associated with kidney function. UMOD has been studied in several cohorts and has shown an association with CKD progression. However, its independent predictive value has not been confirmed in multiple cohort studies or randomized clinical trials, and, thus, it is not considered validated [41,66,67].

a1M (alpha-1-microglobulin) is a biomarker of tubular injury and is associated with CKD progression. a1M has been studied in various cohorts and has shown an association with CKD progression. However, its independent predictive value has not been confirmed in multiple cohort studies or randomized clinical trials, and, thus, it is not considered validated.

MicroRNAs are useful markers in diabetic nephropathy [68].

The CKD273 panel is a set of 273 urinary peptides that are associated with CKD progression. This panel has been shown to predict the progression of diabetic nephropathy, a common cause of CKD. The CKD273 panel has been validated in multiple cohort studies, making it one of the few validated biomarkers for CKD progression [69].

## 6. Prognostic Models

Prognostic models like the Kidney Failure Risk Equation (KFRE) are used to predict CKD progression. These models incorporate variables such as age, sex, eGFR, and urine albumin-to-creatinine ratio. More advanced models include additional laboratory data and biomarkers [70,71,72]. The KFRE is a widely used tool for predicting the risk of kidney failure in patients with CKD. The Chronic Kidney Disease Prognosis Consortium (CKD-PC) has developed risk tools available online for clinical use [73]. These tools incorporate various risk factors and biomarkers to predict the risk of CKD progression and adverse outcomes. The CKD-PC risk tools are based on data from large, diverse cohorts and have been validated in multiple populations [74,75,76]. It should be kept in mind that the risk of death for a cardiovascular event may precede that of end-stage renal failure, especially in late stages of CKD. On the other end, some factors (e.g., age, GFR, urine albumin-to-creatinine ratio) have a major role in all patients. Comorbid conditions (e.g., blood pressure, weight, history of heart failure) play a major prognostic role in some patients, and specific factors (e.g., PLA2R for membranous glomerulopathy or kidney volume for autosomal dominant polycystic kidney disease) in rare cases of CKD.

## 7. How to Reduce Progression

Effective management strategies are essential to slow the progression of CKD and improve patient outcomes. Key interventions include dietary modifications, pharmacological treatments, and surgical options, to reduce the progression rate of CKD.

### 7.1. Dietary Interventions

#### 7.1.1. Low-Protein Diet

A low-protein diet is recommended to slow CKD progression by reducing hyperfiltration and intraglomerular pressure. Studies have shown that protein restriction can reduce glomerular hyperfiltration, metabolic acidosis, and hyperphosphatemia [77,78]. A plant-dominant low-protein diet may also modify the gut microbiome and reduce the generation of uremic toxins.

#### 7.1.2. Sodium Restriction

Sodium restriction is advised to reduce blood pressure, which is crucial in managing CKD. However, the role of diet may be limited, considering the availability of new pharmacological treatments [79].

### 7.2. Pharmacological Interventions

Renin–Angiotensin System Inhibitors:

ACE inhibitors (ACEi) and angiotensin receptor blockers (ARBs) have long been used to manage CKD by reducing blood pressure and proteinuria [80,81,82].

Sodium–Glucose Cotransporter-2 Inhibitors (SGLT2i):

SGLT2 inhibitors, such as dapagliflozin, have shown significant efficacy in slowing CKD progression by reducing hyperfiltration and providing renal protection [83,84,85].

Glucagon-Like Peptide-1 Receptor Agonists (GLP-1 RA):

GLP-1 receptor agonists, like semaglutide, offer potential kidney protective effects by improving glycemic control and reducing inflammation [86,87].

Selective Mineralocorticoid Receptor Antagonists:

Finerenone, a selective mineralocorticoid receptor antagonist, has demonstrated benefits in reducing albuminuria and slowing CKD progression in patients with type 2 diabetes [88].

Endothelin-A Receptor Antagonists:

Endothelin-A receptor antagonists, such as atrasentan, are promising new drugs that target renal fibrosis and inflammation, showing potential in reducing CKD progression [89].

Anti-Fibrosis Agents:

MicroRNAs are emerging as anti-fibrosis agents that can modulate gene expression and reduce renal fibrosis, offering a novel approach to CKD management [81,90].

A combination of drugs may provide benefits for reducing CKD progression [91].

### 7.3. Obesity

Obesity is a major risk factor for CKD progression. Bariatric surgery, such as sleeve gastrectomy, has been shown to significantly reduce weight and improve renal outcomes in obese patients [92].

## 8. Conclusions

CKD is emerging as one of the most common causes of mortality all over the world. Several diseases result in CKD but its progression is controlled by many factors, which have been well investigated. Undertaking measurements of kidney function is still cumbersome, and therefore new simpler techniques should be developed. The list of traditional factors causing the progression of CKD may be changed by new studies. Risk factors of CKD progression are well known, but the role of each factor in an individual patient should be examined. The progression of CKD is influenced by a complex interplay of non-modifiable and modifiable risk factors. Understanding these factors, along with the use of biomarkers and prognostic models, is crucial for predicting and managing CKD progression. The role of diet is now limited, compared to the increasing number of drugs effective in reducing CKD progression. New studies are needed in order to establish when and how they intervene. Prospective studies like the CRIC and CKiD provide valuable data that can inform clinical practice and guide future research.

## 9. Future Directions—Comprehensive Kidney Care: A Holistic Approach

Further experimental and clinical studies on the progression of kidney damage are needed, but an integrated approach is advisable [84,93]. Future nephrology will benefit from embedding research into clinical practice, utilizing electronic health records, implementing personalized medicine approaches, integrating Artificial Intelligence (AI) for data analysis, and creating research networks. Collaboration between nephrologists, AI experts, biostatisticians, engineers, data scientists, and pharmacology experts will accelerate the development of precision drug delivery systems and targeted therapies. Active patient participation in treatment and research programs is a priority, facilitated by mobile applications, patient portals, and online educational resources. The remarkable progress made in medical research and innovation offers exciting kidney disease diagnosis and treatment opportunities. However, inequities in access to basic medical testing, healthcare, medications, and sophisticated diagnostic methods remain significant hurdles. Developing a multidisciplinary, future-ready workforce and advocating for continued investment, collaboration, and the translation of innovations from the laboratory to the clinic are essential to optimize care for all individuals living with or at risk of kidney diseases.

### 9.1. Integrated Kidney Care

Integrated kidney care emphasizes a holistic approach that addresses medical, psychosocial, lifestyle, and preventative dimensions of care. This model fosters collaboration among healthcare professionals, including nephrologists, nurses, dietitians, social workers, pharmacists, and psychologists, to optimize outcomes and improve patient experiences. Key elements include sharing information, coordinating appointments and tests, and ensuring timely access to laboratory testing and specialty consultations. Patient engagement through education and support promotes self-management and shared decision-making, moving away from a paternalistic approach. Effective communication between primary care, nephrology, and other specialty clinics is crucial for success.

### 9.2. Health Information Technology

Health information technology is vital for sharing patient information, facilitating remote monitoring, enabling telemedicine consultations, and enhancing care coordination. However, access to such technology varies significantly by region, demographics, and socioeconomic status, even in high-income countries.

### 9.3. Prevention of Kidney Disease

Population health management strategies are essential for the early identification of individuals at risk of kidney disease. These strategies include education campaigns, regular screening programs, and the promotion of healthy lifestyles. Vulnerable populations, such as socioeconomically disadvantaged groups and minority ethnic groups, are at higher risk and require targeted outreach efforts. Utilizing social media, mobile health apps, and other digital platforms can help strategies to reach a wider audience. Collaborating with community organizations, healthcare providers, and public health agencies can amplify the message and ensure information reaches those who need it most.

### 9.4. Telemedicine and Home-Based Care

Telemedicine enables remote monitoring and consultations, thereby improving access to care, especially for patients in rural or underserved areas. Remote monitoring devices can collect real-time patient data, allowing nephrologists to provide timely interventions and minimize hospital visits. Home dialysis options, such as peritoneal dialysis and home hemodialysis, offer greater flexibility, protection against hospital-linked complications, improved quality of life, and potentially better clinical outcomes. Personalized treatment plans that consider individual patient preferences and circumstances are becoming more common.

### 9.5. Equitable Kidney Care

Addressing health inequities is crucial for equitable kidney care. Inequities are associated with an unequal distribution of risk factors and access to care, high therapeutic costs, and social determinants of health such as poverty, social injustice, and a lack of education. Innovative strategies are needed to tackle barriers to equitable kidney health, including specific communication strategies for minority groups and less privileged populations. Efforts to address sex-dependent health inequities and the under-representation of women in CKD clinical trials are also necessary.

## Figures and Tables

**Table 1 biomedicines-12-02203-t001:** Risk factors of CKD.

**Genetic factors**	*APOL1* gene	Non-modifiable
	Uromodulin	Non-modifiable
**Demographic factors**	Geographical location	Non-modifiable
	Race and ethnicity	Non-modifiable
**Individual factors**	Age	Non-modifiable
	Sex and gender	Non-modifiable
	Birth weight	Non-modifiable
	Familiarity of kidney disease	Non-modifiable
	Pregnancy	Non-modifiable
**Cardiovascular factors**	Hypertension	Modifiable
	Heart failure	Modifiable
	Atrial fibrillation	Modifiable
**Metabolic factors**	FGF2	Modifiable
	Serum bicarbonate	Modifiable
	Parathyroid hormone	Modifiable
	Serum hemoglobin	Modifiable
	Serum phosphate	Modifiable
	Serum uric acid	Modifiable
	Urinary oxalate	Modifiable
	Dyslipidemia	Modifiable
**Lifestyle factors**	Diet	Modifiable
	Lack of physical activity	Modifiable
	Smoking	Modifiable
	Socioeconomic status	Modifiable
**Other factors**	Acute kidney injury	Modifiable
	Renal fibrosis	Modifiable
	Gut–renal axis	Modifiable
	Proteinuria	Modifiable
	Inflammatory states	Modifiable
	Proximal tubule injury	Non-modifiable

**Table 2 biomedicines-12-02203-t002:** Biomarkers of CKD.

Biomarker	Validated	Comments
TNFR1	No	Associated with inflammation and kidney damage, not validated in multiple cohort studies or RCTs.
TNFR2	No	Similar to TNFR1, associated with inflammation, not validated in multiple cohort studies or RCTs.
suPAR	No	Linked to increased risk of CKD progression, not validated in multiple cohort studies or RCTs.
KIM-1	No	Biomarker of tubular injury, not validated in multiple cohort studies or RCTs.
NGAL	No	Biomarker of acute kidney injury, not validated in multiple cohort studies or RCTs.
L-FABP	No	Associated with tubular injury, not validated in multiple cohort studies or RCTs.
UMOD	No	Linked to kidney function, not validated in multiple cohort studies or RCTs.
a1M	No	Biomarker of tubular injury, not validated in multiple cohort studies or RCTs.
CKD273 Panel	Yes	Validated in multiple cohort studies, predicts progression of diabetic nephropathy.

RCT: Randomized Controlled Studies.

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
