# Peer review of "Chronic Kidney Disease Progression—A Challenge"

_biomedicines, 2024, doi:10.3390/biomedicines12102203_

Round 1

Reviewer 1 Report

Comments and Suggestions for Authors

The topic of this paper (chronic kidney disease progression) is an important one and the authors provide a very well written review. I especially appreciated the fact that the review is comprehensive and well structured, offering a didactic and yet modern presentation of the subject.

Lines 192-193 The reference (Thomas Dienemann et al., BMC Nephrol. 2016 Sep 2;17(1):121. doi: 10.1186/s12882-016-0335-2) should be removed. It is already cited (number 26).

I believe a section or at least a paragraph dedicated to the costs of chronic kidney disease progression costs (and maybe also progression prevention costs) would be an addition interesting for the readers and would make the analysis complete.

Also, please explain how the articles included in the review were selected.

Please review the references. reference 1 has no authors listed. Also, tehre is a problem with reference 44. It starts with an initial.

Reviewer 2 Report

Comments and Suggestions for Authors

Dear authors,

Dear authors

After analyzing the article my concern is that it is to basic for the special issue about CKD.  It is challenging to make a prognosis regarding CKD development.  In the introduction part row 30 the authors are writing about CKD as a progressive condition where the reduction of renal function continues independently of the initial cause of damage or external factors such as infection, inflamations or toxins but there are also other causes leading to CKD in pediatric population the main cause is CAKUT. Chapter 2 how to measure renal function: cathionized ferritin enhanced MRI and inulin clearance are not mentioned and those are important new methods. 2.2 estimates of GFR it is important to describe why race is so important in estimating GFR. Very nice is the concept of renal reserve but is not well documented.

2.4 Current classification of CKD the authors are using KDIGO 2021 guidelines but there is a new 2024 KDIGO guideline for CKD. Also the biomarkers for CKD are not well discribed with novel refferences.

Round 2

Reviewer 1 Report

Comments and Suggestions for Authors

Thank you for answering all the queries. All the required modifications were performed in a satisfactory manner. The article is worthy to be published in the current form.

Reviewer 2 Report

Comments and Suggestions for Authors

The article is now much better after the suggestions I made and which were applied but I still think that the article doesn't bring novel information and does not fit to the profile of the journal. It is a synthesis of everything that was known until now and does not discover anything new.